# The Impact of Magnesium–Aluminum-Layered Double Hydroxide-Based Polyvinyl Alcohol Coated on Magnetite on the Preparation of Core-Shell Nanoparticles as a Drug Delivery Agent

**DOI:** 10.3390/ijms20153764

**Published:** 2019-08-01

**Authors:** Mona Ebadi, Kalaivani Buskaran, Bullo Saifullah, Sharida Fakurazi, Mohd Zobir Hussein

**Affiliations:** 1Materials Synthesis and Characterization Laboratory, Institute of Advanced Technology (ITMA), Universiti Putra Malaysia, Serdang 43400, Selangor, Malaysia; 2Laboratory for Vaccine and Immunotherapeutic, Institute of Biosciences, Universiti Putra Malaysia, Serdang 43400, Selangor, Malaysia; 3Institute of Advanced Research Studies in Chemical Sciences, University of Sindh, Hosho Raod Jamshoro Sindh, Jamshoro 76080, Pakistan; 4Department of Human Anatomy, Faculty of Medicine & Health Sciences, Universiti Putra Malaysia, Serdang 43400, Malaysia

**Keywords:** nanodrug delivery, liver cancer, polyvinyl alcohol, magnetite, 5-fluorouracil

## Abstract

One of the current developments in drug research is the controlled release formulation of drugs, which can be released in a controlled manner at a specific target in the body. Due to the diverse physical and chemical properties of various drugs, a smart drug delivery system is highly sought after. The present study aimed to develop a novel drug delivery system using magnetite nanoparticles as the core and coated with polyvinyl alcohol (PVA), a drug 5-fluorouracil (5FU) and Mg–Al-layered double hydroxide (MLDH) for the formation of FPVA-FU-MLDH nanoparticles. The existence of the coated nanoparticles was supported by various physico-chemical analyses. In addition, the drug content, kinetics, and mechanism of drug release also were studied. 5-fluorouracil (5FU) was found to be released in a controlled manner from the nanoparticles at pH = 4.8 (representing the cancerous cellular environment) and pH = 7.4 (representing the blood environment), governed by pseudo-second-order kinetics. The cytotoxicity study revealed that the anticancer delivery system of FPVA-FU-MLDH nanoparticles showed much better anticancer activity than the free drug, 5FU, against liver cancer and HepG2 cells, and at the same time, it was found to be less toxic to the normal fibroblast 3T3 cells.

## 1. Introduction

Cancer is one of the most hazardous illnesses, and the main reason for death in the world [1,2]. In 2000, about 22 million patients were living with cancer [3]. Liver cancer is the most common cancer-related death and is usually treated with surgical procedures, radiation, and chemotherapy [4].

The development of new methods for diagnosis and treatment of cancer is important to increase patients’ life survival. In recent years, nanotechnology and nanoscience have enabled early detection, accurate diagnosis, and more effective delivery of anticancer drugs to cancer cells [1,2,3,5]. One of the main challenges in the treatment of cancer is engineering drug carriers for simultaneous specific targeting and drug unloading [6]. Nanomedicine is providing opportunities to create novel nanoparticle formulations with better therapeutic outcomes in cancer therapy. 

There is a risk of killing healthy cells or damaging normal tissue in cancer treatments. This problem can be solved by delivering the drug directly to the tumor cells and target site using organic or inorganic nanoparticles [7,8,9,10,11].

Recently, magnetite nanoparticles (FNPs) consisting of small crystallites have emerged as a new generation of targeted drug delivery and therapy. Their physico-chemical properties can be manipulated so that they cause minimum side effects, show increased biodegradability, are better guided to target sites by an external magnetic field, are degraded into nontoxic ions in vivo, and have a long blood retention time with low toxicity [12,13,14]. However, one of the major problems with magnetic nanoparticles is that their stability is relatively low. One of the methods to solve this problem is by the formation of core-shell drug carriers using Fe_3_O_4_ as the core and functional materials as the shell. This could increase the stability of the drug as well as the ability to control the release of the drug [6]. In addition, it has been found that even minor discrepancies in the iron oxide nanoparticles (FNPs) composition as the carrier can result in a severe impact on the cellular response to FNPs [15].

The properties of FNPs can be modified by adding a coating to their surface. Due to their unique properties, they are able to carry a considerable amount of drug on their surfaces. The polymer is usually used in this case due to its amazing and unique properties [16,17]. Magnetite coated with polymers such as polyvinyl alcohol (PVA) resulted in a stable, biocompatible compound with an increased circulatory half-life, high capacity drug loading, high ability to target drug, and reduced toxic side effects on healthy tissues. Magnetite nanoparticles have also been stabilized by polymer to decrease the size distribution of the particles by avoiding agglomeration [11]. Polyvinyl alcohol (PVA) has a simple chemical structure and is suitable to be used for different areas of medical applications, because of its excellent chemical properties, nontoxicity, active delivery to targeted cells, and capacity to increase the water solubility for many low solubility molecules. In addition, 5-fluorouracil as an anticancer drug can be attached to the targeting ligands for tumor-specific drug delivery [18].

Coated-magnetite can be synthesized where magnetite as a core is coated with magnesium-aluminum–layered double hydroxide as a shell. Layered double hydroxide is biocompatible and it has a two-dimensional (2D) layered structure consisting of cationic brucite and exchangeable interlayer anions. This positively charged, inorganic, 2D material has known for many years as a multifunctional 2D nanomaterial, and is widely used in biomedical applications, especially as a carrier to deliver drugs [19,20]. This material has exchangeable anion capability, and its drug nanocomposites, in which the drug is incorporated between the 2D LDH interlayers, have controlled release properties. All drug carriers should be nontoxic and have high drug-loading efficiency, demonstrating the improved therapeutic effect, protecting the intercalated drug from degradation, and decreasing the toxicity to healthy cells [21,22,23]. Magnetic core-shell nanoparticles with ideal shell materials are one of the most effective strategies to prevent the magnetic core from aggregation and as a platform for drug delivery application [24].

Additionally, a suitable target agent of the anticancer drug must be carefully selected, so that it will help in delivering the drug to the target cells, i.e., the tumor sites. Anticancer drugs should be biocompatible and induce dose-limiting toxicities. Anticancer drugs can be soluble, adsorbed, attached, or encapsulated into nanocomposites, which could sustain release the drugs over a longer time. 5-fluorouracil (5FU), an anti-metabolite and pyrimidine analog, has been extensively and successfully used to treat breast, stomach, pancreatic, and liver cancers [25]. 5FU came into medical use in 1962; however, common side effects attached to it, such as inflammation of the mouth, loss of appetite, low blood cell counts, hair loss, and inflammation of the skin are the drawbacks, and it is hoped that these drawbacks can be addressed using this core-shell targeted drug delivery approach [26,27].

In the drug delivery system, targeted polymeric nanocore–shells have attracted great attention due to their better properties and wide applications. Superparamagnetic properties are necessary to control the residence time of a drug in a target area by an external magnetic field [28]. Whenever the external magnetic field is removed, magnetization disappears and the nanoparticles are accumulated in capillary, organs, and cells. Polymeric nanocore–shells have been used to increase drug solubility, safety, and, more importantly, increase delivery efficiency [29,30].

Previous works have shown that the cell’s mechanical and structural support system, the extracellular matrix (ECM), can be exploited for the diagnosis and treatment of various health problems [31]. These biomolecules interact with nanoparticles in all biological processes, and are therefore useful for nanomedicine applications [31]. Moreover, toxicity data are important in new materials and this can only be obtained after the toxicity evaluation. The concentration addition (CA) and independent action (IA) are important for assessing a mixture’s toxicity of chemicals [32].

Previous works have shown that novel nanocomposites composed of metal-organic frameworks (MOFs) or graphene oxide (GO) can be attached to the surface of Fe_3_O_4_ and subsequently used as ibuprofen nanocarriers. They found that the nanocomposites were stable, had high adsorption capacity, and were more efficient in loading and release of the drug [24]. In addition, magnetic drug nanocarriers containing γ-Fe_2_O_3_ coated with porous silica and modified by poly (2-dimethylamino ethyl methacrylate) along with doxorubicin have been synthesized as a chemotherapy medication [6]. Moreover, a new smart drug carrier, composed of γ-Fe_2_O_3_/p-silica containing a γ-Fe_2_O_3_ as a core and porous silica as a shell [33], was also synthesized. It was found that porous silica could increase drug loading with controlled released property [6,33].

In this paper, we describe our work on the synthesis and characterization of core-shell nanoparticles for cancer therapy. This is based on a superparamagnetic material, a Fe_3_O_4_ core, coated on the surface of the core with polymer, loaded with drug together with Mg-Al-LDH as a shell to provide stability with less toxicity and with better-targeted delivery properties. In addition, the core-shell nanocarriers also were loaded with an anti-cancer drug, 5-fluorouracil.

## 2. Results and Discussion

### 2.1. X-Ray Diffraction

The effects of coating on the properties of the synthesized core-shell nanoparticle were investigated via X-ray diffraction study. X-ray diffraction patterns for the pure magnetite nanoparticles, FPVA-FU-MLDH, pure polyvinyl alcohol, Mg-Al-LDH, and 5-fluorouracil are presented in Figure 1. The characteristic peaks between 2–80° in the XRD patterns correspond to the (220), (311), (400), (422), (511), and (440) reflections, representing the cubic and inverse spinal structure of the magnetite crystals (JCPDS reference number 19-629) [30]. The X-ray diffraction pattern at 2θ angles of 19.5° is due to the presence of PVA polymer on the surface of the magnetite nanoparticles [34]. The three main peaks correspond to diffraction at the 2θ positions of 11.5°, 23.2°, and 34.8° for (003), (006), and (009) are respectively related to the pure Mg-Al-LDH [35]. The diffractogram in Figure 1D indicates a sharp peak of 5-fluorouracil, the anticancer drug, at 28.44°. As observed in Figure 2, the peak intensity decreased after the nanoparticles were coated with other compounds, owing to the interaction between the nanoparticles. Impurity peaks were not observed in the XRD pattern, indicating that the synthesized particles were of good purity and the coating process did not change their phase. 

Using Sherrer’s equation (D = Kλ/βcosθ), the mean crystallite size (D) was estimated, where K is the Debye–Scherrer constant (0.94), λ is the X-ray wavelength for CuK_α_ (0.154 Å), β is the peak width of half-maximum, and θ is the Bragg angle. Using the X-ray results and the Debye–Scherrer equation, the size of the core-shell nanoparticle was found to be in the range of 8–9 nm. 

### 2.2. Fourier Transform Infrared Spectra

The Fourier transform infrared spectra (FTIR) are shown in Figure 3, and were recorded to understand the chemical and molecular interactions between polymer, drug, LDH nanocarrier, and the magnetite nanoparticles. The absorption band recorded at 560 cm^−1^ (Figure 3A) was assigned to the stretching vibration of the Fe–O group. The band related to the Fe–O was shifted to 528 and 513 cm^−1^ in FPVA and FPVA-FU-MLDH, respectively. The wavenumber shifts indicate that some of the interactions have taken place. As can be seen, in the FPVA nanocomposite (Figure 3C,F), C–H stretching and bending vibrations were observed at 2760 and 2360 cm^−1^, and bending vibrations at 1415 and 1466 cm^−1^, which were assigned to C–H stretching. The bands at 1236 and 1027 cm^−1^, which appear in Figure 3C and at 1225 and 1015 cm^−1^ in FPVA-FU-MLDH were associated with the C–O–C stretching vibration and –C–O group, respectively. This confirmed the presence of PVA in the magnetite core-shell nanoparticles. For the MLDH FTIR spectrum (Figure 3D), the O–H bending vibration band was observed at 3416 cm^−1^ [36,37]. It can be observed that this band was shifted to 3398 cm^−1^ due to the formation of a bond between LDH and FPVA–drug nanoparticles. 

Bands at 1344 and 966 cm^−1^ were due to the nitrate group (NO_3_^-^ stretching vibration) [38] and O–M–O lattice vibration [39], respectively. This band was not present in the final sample due to the fact that in the bonding process, the nitrate anion was removed from the LDH interlayers. The shifts occurred in bond absorption is as a result of bonding that took place between the guest and the host. On the other hand, an H–O–H bending vibration could be seen at 1636 cm^−1^ and another band at 1344 cm^−1^, which were ascribed to the stretching modes of C–O and were shifted to 1622 and 1343 cm^−1^ in the synthesized sample (Figure 3F). Furthermore, the FTIR spectrum for FPVA-FU-MLDH depicted a vibrational band at 1533 cm^−1^, which may have been due to the amine N−H stretching band. This suggests that PVA and 5-fluorouracil were successfully coated onto the magnetite nanoparticles, in parallel with the XRD findings mentioned earlier. 

### 2.3. Magnetic Properties 

Superparamagnetic property is important for magnetic targeting of carriers in drug delivery; therefore, the magnetic properties of FNPs, FPVA (magnetite coated with polyvinyl alcohol), and FPVA-FU-MLDH (magnetite coated with PVA, co-coated with MLDH and loaded with 5FU) were investigated at room temperature by a vibrating sample magnetometer (VSM). The values for saturation magnetization (M_s_), remnant magnetization (M_r_), and high coercivity (H_ci_) are shown in Table 1. Figure 4 shows that the value of saturation magnetization for the magnetite nanoparticles was about 80 emu/g, while for FPVA and core-shell nanoparticle it was about 49 and 11 emu/g, respectively. This high degree of superparamagnetic characteristics of the nanoparticles is highly desirable for medical applications. 

Reduction of the saturation magnetization in FPVA was related to the exchange of electrons between the surfaces of magnetite with a coating of PVA polymer. This observation in the resulting synthesized sample was presumably due to the increase of crystallinity due to the effect of the surface of magnetite nanoparticles attributed to the coated material and impurities. In both the synthesized samples, remanence and coercivity properties were observed, which confirmed their superparamagnetic property, meaning that after removal of the magnetic field, they did not retain any magnetism. 

### 2.4. Thermogravimetric Analyses

Thermal gravimetric and differential thermogravimetric (TGA/DTG) analyses are thermal degradation analysis techniques made of the pure materials and the nanocomposites, where the percentage weight loss of a sample was measured over temperature changes. This measurement indicates changes in mass through thermal decomposition, dehydration, and thermal oxidation of a sample, usually between 25–1000 °C. The nanoparticles and the pristine polyvinyl alcohol, 5-fluorouracil, Mg-Al-LDH, and FPVA-FU-MLDH-coated nanoparticles after drug intercalation under atmospheric conditions were tested, and the thermal decompositions of them are depicted in Figure 5. As shown in Figure 5A, for free polyvinyl alcohol, the thermogravimetric analysis indicated two stages of weight loss. The peak at 265 °C corresponds to a weight loss of 76.7%, which was attributed to polyvinyl alcohol side chain elimination, and the second peak at 419 °С with 9% mass loss is due to the depreciation of the main PVA chain [40,41,42,43]. Figure 5B shows the thermal curves and demonstrates a sharp mass reduction step at 329 °C, indicating 89.3% weight loss, which could be due to the decomposition of 5-fluorouracil. Thermal decomposition at 180–370 °C is due to the decomposition of 5FU. This weight loss observed may be due to homolytic rupture of N–H bonds and ring secession. The thermograms of FPVA-FU-MLDH core-shell nanoparticles shown in Figure 5C display that the mass loss appeared between 30 °C and 900 °C and shows three thermal decomposition peaks. The first stage of weight loss started at 45 °C with 1.8% weight loss, corresponding to the removal of residual water.

The onset of the degradation of FPVA-FU-MLDH core-shell nanoparticles occurred in five stages of mass loss at 157 °C (6%), 282 °C (7.7%), 548 °C (1.2%), 777 °C (5%), and 830 °C (4%). The second weight loss was representative of the loss of the chain of the iron atom [44]. The major decomposition at 282 °C that corresponds to 7.7% weight loss is due to the thermal decomposition of nitrate ions of the MLDH nanocarrier [36]. The weight loss between 500–800 °C involved the removal of interlayer anion decomposition and the collapse of the layered structure. The sharp peak at 830 °C with 4% mass loss is associated with the complete decomposition of the nanoparticles that indicates better thermal stability owing to the electrostatic attraction bonding between the negatively charged groups and the positively charged LDH interlayer surfaces. This demonstrates the existence of polyvinyl alcohol, MLDH, and 5FU together on the surface of the magnetite nanoparticles.

### 2.5. Particle Size Distribution by the Dynamic Light Scattering Technique 

The particle size distribution, which was obtained using the dynamic light scattering (DLS) technique, is shown in Figure 6A. The 100% cumulative distribution is revealed in Figure 6B. The average particle size for core-shell nanoparticles was found to be very broad, in the range of 43 and 105 nm. This figure shows that around 60% of the nanoparticles had a hydrodynamic diameter size of 68 nm, which is suitable for drug delivery application. The result is similar to the one obtained using the transmission electron microscope (TEM). However, due to the high possibility of aggregation of the nanoparticles, the diameters obtained by the TEM method were more reliable in representing their real sizes. 

### 2.6. Particle Size Distribution by the Transmission Electron Microscopy Technique

The transmission electron microscopic (TEM) technique is a very powerful tool to investigate the finest details of the internal structure of nanoparticles. Transmission electron microscopy study of magnetite nanoparticles and core-shell nanoparticles was performed to obtain information about the general morphology, shape, size, size distribution, structural features and uniformity of the nanoparticles synthesized under optimum conditions. The particle size distributions of FNPs and FPVA-FU-MLDH were studied before and after the addition of coating materials covering the surface, and were calculated using the image analysis software from at least 100 particles chosen at random. The TEM images are presented in Figure 7. Figure 7A is the micrograph of the magnetite nanoparticles showing the mean diameter of the magnetite nanoparticles, which is in the range of 19 nm with the monomodal distribution (Figure 7C). 

Figure 7B indicates the TEM micrographs of the core-shell nanoparticles, and the core-shell structure can be seen with the coating layers also visible. As indicated by the size distribution histogram, after coating, the size of the core-shell nanoparticles appeared to be larger, increased to 31 nm in diameter (Figure 7D). This suggests that the core-shell nanoparticles were enlarged because the magnetite nanoparticles were agglomerated over time, due to high surface energy interaction affinity [45,46] and in addition, the drug also was intercalated into the LDH. As expected, the prepared nanoparticles are almost spherical in shape, with fairly broader particle size distribution compared to the magnetite nanoparticles.

### 2.7. Loading and Release Behavior of 5-Fluorouracil

The release behavior of 5-fluorouracil, an anticancer drug from the FPVA-FU-MLDH core-shell nanoparticles was investigated using phosphate-buffered solutions (PBS) at pH 7.4 and 4.8 and UV-vis spectroscopy. Figure 8 shows 5FU release profiles from the physical mixture of the drug and the nanoparticles. The drug can be seen to be rapidly released at the beginning; 87% and 81% release were completed within 4 and 7 min into the phosphate-buffered solution at pH 4.8 and pH 7.4, respectively. The fast release of 5FU was presumably due to the dissolution of the nanoparticles in the acidic environment and the breaking of the bonds between the polymer and the drug. The slower release in the alkaline pH was related to the exchange of anions in the nanocarrier with the anionic moieties in the solution, and lower electrostatic attraction between the 5-fluorouracil anions and the FNPs-PVA-MLDH. The 5FU release rate from FPVA-FU-MLDH was obviously slower and sustained as compared to its physical mixture. This was due to improved chemical interaction between the drug and FPVA-MLDH nanocomposite, and the controlled release through interaction between the negatively-charged drug and positively-charged nanocarriers.

The release profile of 5FU from FPVA-FU-MLDH nanoparticles, shown in Figure 9, indicates that the maximum percentages of the release rate reached around 93% within 56 h when they were exposed to PBS at pH 4.8, compared to only 76% by 127 h at pH 7.4. The difference in the release mechanism is presumably due to the pH-dependent properties of the nanoparticles. The release rate at pH 4.8 increased at pH 7.4. Under an acidic environment, FPVA-FU-MLDH nanoparticles are unstable. However, the drug release process was slower and more stable under pH 7.4, and release occurred through the ion exchange process between the drug anions and the negatively-charged ions available in the buffer solution.

### 2.8. Kinetics of 5-Fluorouracil Release from the Nanoparticles

The release kinetics of 5-FU from the FPVA-FU-LDH nanocomposites at pH 7.4 and 4.8 can be described using several kinetic models: first-order (Equation (1)), pseudo-second order (Equation (2)), and parabolic diffusion (Equation (3)):ln (q_e_ − q_t_) = ln q_e_ − k_t_(1)
t/q_t_ = 1/k_2_ q_e_^2^ + t/q_e_(2)
(1 − M_t_/M_o_)/t = kt ^−0.5^ + b(3)
where, in these equations, the q_e_ and q_t_ are the equilibrium release rate and the release rate at time t, respectively; k is a constant corresponding to release rate; and M_o_ and M_t_ are the drug content remaining in the nanocomposite and the nanocarrier at release times 0 and t, respectively. 

The kinetic parameters of the amount of the drug released using the ultraviolet-visible spectroscopy were investigated. The plots of the fitting of t/q_t_ against time for the release of the drug from the nanoparticles are displayed in Figure 10, and the correlation coefficient (R^2^), percentage of saturation release, the rate constant k, and the half-life of the release (t_1/2_) are given in Table 2. The data for the release of the active drug (5FU) in three kinetic models indicate that the pseudo-second-order model was the best model to describe the release kinetic behavior of 5FU from the nanoparticles for both pH 7.4 and 4.8. Based on Figure 10 and Table 2, the fitting to the pseudo-second-order model resulted in a correlation coefficient, R^2^, of 0.9996, and a rate constant (k) value of 4.4 × 10^−4^ mg/min fit better for a phosphate-buffered solution at pH 4.8. The release of 5FU from FPVA-FU-MLDH at pH 7.4 was also governed by the pseudo-second-order model with a correlation coefficient of 0.9986. 

### 2.9. In Vitro Bioassay

All the cytotoxicity assays were carried out in triplicate, and the standard deviations were calculated and are incorporated in the respective bar graphs. For the calculation of IC_50_, we put the x-and y-axis values in the graph and converted the x-axis values (conc.) to their log values, followed by nonlinear regression (curve fit) under the XY analysis to obtain y = an (x) + b equation, from which the inhibition IC_50_ was calculated.

#### 2.9.1. Cytotoxicity Studies on Normal Fibroblast 3T3 Cells

Cytotoxicity studies were conducted by treating the magnetite (FNPs), FPVA, FPVA-MLDH, 5-fluorouracil and the core-shell nanoparticles (FPVA-FU-MLDH) with normal fibroblast 3T3 cells using the colorimetric assay. The 3T3 cells were employed to confirm the nontoxic temperament of the nanoparticles. Various gradient concentrations of the samples were incubated for a maximum of 72 h with the 3T3 cells. Cell viability was determined using the standard 3-(4,5-Dimethylthiazol-2-yl)-2,5-Diphenyltetrazolium Bromide (MTT) assay protocol.

Figure 11 shows the percentage of cell viability of the 3T3 cells after 72 h incubation for all the samples. The magnetite (FNPs), FPVA, FPVA-MLDH, 5FU and the nanoparticles (FPVA-FU-MLDH) were found to be biocompatible and nontoxic; the cell viability was found to be more than 70% after 72 h of incubation. This suggests that the designed anticancer nanoparticle formulation is biocompatible with normal cells and would be very useful for targeting cancer cells without damaging/harming the normal tissues. The statistical analysis using the analysis of variance (ANOVA) and Duncan’s multiple range tests revealed that no significant difference was found among the sample groups at individual concentrations.

#### 2.9.2. Anticancer Action against Liver Cancer Cells, HepG2 

The MTT assay for assessing cell metabolic viability has been described as a convenient and reliable method for the detection of toxicity. The colorimetric assay reflects the number of viable cells that are still present. For the study of the anticancer activity and effectiveness of the synthesized nanoparticles, including the bare magnetite (FNPs), FPVA, FPVA-MLDH, 5FU and the nanoparticles (FPVA-FU-MLDH), the samples were treated with liver hepatic cancer cells, HepG2. Different concentrations of the samples (1.25, 3.125, 6.25, 12.5, 25, 50 and 100 μg/mL) were incubated with HepG2 for hepatocyte enzymes for 72 h, and the viability of cells was measured by MTT assay protocol. Figure 12 demonstrates the efficacy of HepG2 cells following exposure to the nanoparticles after 72 h incubation with increasing concentrations on similar cells. The empty carriers, FNPs, FPVA, and FPVA-MLDH did not show any inhibitory action against liver cancer cells. 

The IC_50_ values of the pure drug (5FU) and FPVA-FU-MLDH against liver cancer cells, HepG2, were found to be 32.73 and 21.53 µg/mL, respectively. The effective IC_50_, which is the actual amount of 5FU present in IC_50_ of the anticancer nanoparticles was calculated from the loading percentage of the drug, 5FU, which was 68.4%, as determined by HPLC analysis. These results suggest that the nanoparticles had much better anticancer activity compared to the free drug (5FU). Statistical analysis was carried out using ANOVA and Duncan’s multiple range test and the SPSS software. There was a significant difference between the empty carrier, magnetite (FNPs), FPVA, FPVA-MLDH, free 5FU and FPVA-FU-MLDH-treated groups. There was a significant difference between them at concentrations of 6.25 to 25μg/mL with *P* values of <0.5. A difference was also observed at concentrations of 12.5, 25, 50, and 100 μg/mL.

The samples; magnetite (FNPs), FPVA, FPVA-MLDH, 5FU, and FPVA-FU-MLDH show an anticancer effect for this cell line in a dose-dependent manner, where after 72 h, the dose-dependence was decreased and the cytotoxicity was increased with the increased of the concentration. The half-maximal inhibitory concentration value (IC_50_) of all the samples is given in Table 3. The IC_50_ values of the nanoparticle determined based on the percentage of drug loading clearly indicate that the nanoparticles had a better anticancer effect than the bare drug. 

## 3. Materials and Methods 

### 3.1. Materials and Method

Ferrous chloride tetrahydrate (FeCl_2_·4H_2_O) and ferric chloride hexahydrate (FeCl_3_·6H_2_O) with 99% purity were supplied by Merck (Darmstadt, Germany). Ammonia solution (25%) was obtained from Scharlau (Barcelona, Spain). In order to coat the magnetite nanoparticles with polymer, polyvinyl alcohol (PVA) with an average M.W. of 6000 was used, which was sourced from Acros Organics (Morris plains, NJ, USA). Aluminum nitrate (Al(NO_3_)_3_·9H_2_O) (98.5% purity) and magnesium nitrate (Mg(NO_3_)_2_·6H_2_O) (99% purity) were acquired from ChemAR (Kielce, Poland). 5-fluorouracil (C_4_H_3_FN_2_O_2_) was purchased from AKSci (Union, CA, USA) at 98% purity. Dimethyl sulfoxide ((CH_3_)_2_SO) of 99% purity was also used as a solvent for the drug and was obtained from Sigma Aldrich (St Louis, MO, USA). Deionized water (18.2 M Ωcm^−1^) was used in the preparation of all the aqueous solution. 

### 3.2. Experimental Section


**Syntheses of magnetite nanoparticles coated with polyvinyl alcohol**


The magnetite nanoparticles were synthesized via co-precipitation method. First, 2.43 g of iron (II) chloride tetrahydrate (FeCl_2_.4H_2_O), 0.99 g of iron (III) chloride hexahydrate (FeCl_3_.6H_2_O), 6 mL of ammonia (25% by mass), and deionized water was made up to a volume of 100 mL. The solution was exposed to ultrasonic irradiation at room temperature, and then centrifuged and washed three times. The surface-coated FNPs were obtained using 50 mL deionized water added to 2 g of polyvinyl alcohol (PVA) and the resulting material was washed several times. The washed precipitates obtained were then dispersed in a mixture of 2% PVA. The nanoparticles obtained were placed in an autoclave at 200 °C for 24 h to remove the uncoated PVA, and the resulting black precipitates were washed at least three times with deionized water.


**Drug loading process**


For determination of loading of 5-fluorouracil on FPVA, 3 g of 5FU was dissolved in dimethyl sulfoxide containing 3 g of coated magnetite nanoparticles (FPVA), and the mixture was stirred for 24 h, and the resulting FPVA-FU nanoparticles were obtained. 


**Preparation of FPVA-FU-MLDH**


The Mg–Al-layered double hydroxide nanocarrier (MLDH) was prepared by the co-precipitation method. The solutions of Mg(NO_3_)_2_·6H_2_O and Al(NO_3_)_3_·9H_2_O were dissolved in distilled water. Then, 0.2 M NaOH was added dropwise under mechanical stirring. The pH of the solution was kept at 9–10. After magnetically stirring, the black mixture was centrifuged to obtain the FPVA-FU-MLDH (5FU-loaded, PVA-coated, and MLDH co-coated on magnetite nanoparticles). They were washed two times and dried in an oven at 40 °C.

### 3.3. Instrumentation

X-ray diffraction was used to determine the crystal structure of the samples in the range of 2–80°, using a Shimadzu XRD 6000 diffractometer, Japan, with CuK_α_ radiation (λ = 1.5406 Å) at 40 kV and 30 mA. Their morphology, particle size, and particle size distribution were determined using a transmission electron microscope (TEM, Hitachi H-7100, Tokyo, Japan) at an accelerating voltage of 100 kV, and the particle size and particle size distribution were determined using the image analysis software (UTHSCSA Image Tool V.6, Lyngby, Danmark). The thermal behavior of the core-shell nanoparticles with MLDH was studied using the thermogravimetric and differential thermogravimetry (TGA/DTG) analyses using a Mettler-Toledo instrument (Greifensee, Switzerland) in the range of 20–1000 °C at a heating rate of 10 °C per minute. The Fourier transform infrared (FTIR) spectra were obtained using a Thermo Nicolet 6700 (AEM, Madison WI, USA) with 0.09 cm^−1^ resolution, using the potassium bromide disk method, at 500–4000 cm^−1^. The magnetic properties were characterized using a vibrating sample magnetometer (VSM, Lakeshore 7404, Westerville, OH, USA).

### 3.4. Cell Culture and MTT Cell Viability Assays

In cell viability assays, normal, healthy human fibroblast (3T3) and human hepatocellular carcinoma cells (HepG2) purchased from ATCC (Manassas, VA, USA) were used, and these cells were grown in the Roswell Park Memorial Institute (RPMI) on 1640 medium (Nacalai Tesque, Kyoto, Japan), supplemented with 10% fetal bovine albumin (Sigma-Aldrich, St. Louis, MO, USA) and 1% antibiotics, containing 10,000 units/mL penicillin and 10,000 μg/mL streptomycin (Nacalai Tesque, Kyoto, Japan). Cells were maintained and incubated in humidified 5% carbon dioxide/95% air at 37 °C. Cell layers were harvested using 0.25% trypsin/1 mM EDTA (Nacalai Tesque, Kyoto, Japan) and seeded in 96 well tissue culture plates at 1.0 × 10^4^ cells/well for 24 h in an incubator to reach 90% confluence for treatment. 

A methyl thiazole tetrazolium (MTT)-based assay was carried out to determine cell viability and cytotoxicity. Magnetite nanoparticles (FNPs), FPVA, FPVA-MLDH, 5-fluorouracil, and FPVA-FU-MLDH nanoparticles stock solutions were prepared by dissolving the compound in 1:1 of dimethyl sulfoxide (0.1%) and Roswell Park Memorial Institute (RPMI). The mixture was then further diluted in the same medium to produce various final concentrations, ranging from 1.25 to 100 μg/mL. Once the cells were attached to the respective wells after 24 h, the test compounds were added until a final volume of 100 μL well was reached. After 72 h of incubation, 10 μL of MTT solution (5 mg/mL in PBS) was added to each well and further incubated for 3 h before being aspirated. Subsequently, 100 μL of dimethyl sulfoxide was added per well in the dark and at room temperature in order to dissolve the purple formazan salt. The intensity of the purple formazan solution, which reflects cell growth, was subsequently measured at a wavelength of 570 nm using a microplate reader (BioTek LE800, Winooski, VT, USA).

## 4. Conclusions

In this study, magnetite nanoparticles coated with polymer, drug and LDH were prepared using the co-precipitation method. The mean dimensions of the uncoated and coated magnetite nanoparticles were found to be around 19 and 31 nm, respectively. The thermal stability of the coated magnetite was higher compared to its uncoated counterpart. The active agent release from the coated magnetite nanoparticles occurred by an anion exchange process through a controlled method, and indicated that the coated magnetite nanoparticles may be used as a controlled release formulation. Further, the results showed that the release behavior of 5-fluorouracil from its nanoparticles, FPVA-FU-MLDH, in a buffered solution of pH 4.8 was more sustained than at pH 7.4. In in vitro bioassay, the study indicated that FPVA-FU-MLDH core-shell nanoparticles are a good candidate to be used as a drug delivery system for cancer treatment.

## Figures and Tables

**Figure 1 ijms-20-03764-f001:**
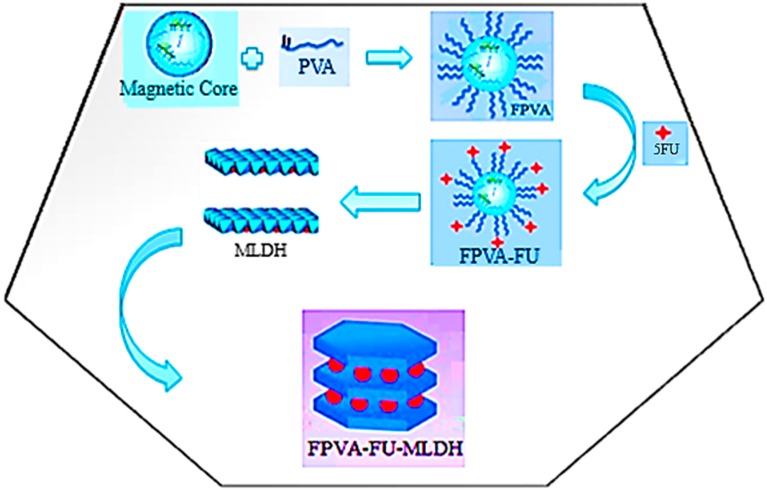
Schematic representation of the core-shell structure of magnetic nanoparticles.

**Figure 2 ijms-20-03764-f002:**
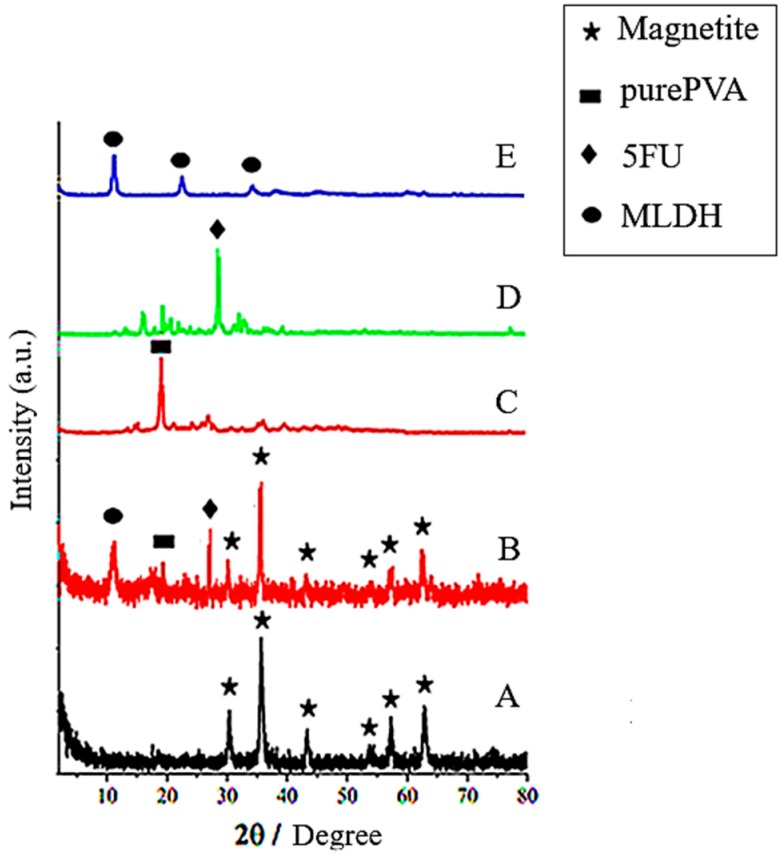
X-ray diffraction patterns for (**A**) Magnetite nanoparticles; (**B**) the core-shell nanoparticles (FPVA-FU-MLDH); (**C**) pure PVA; (**D**) 5-fluorouracil; (**E**) pure Mg-Al-LDH (MLDH).

**Figure 3 ijms-20-03764-f003:**
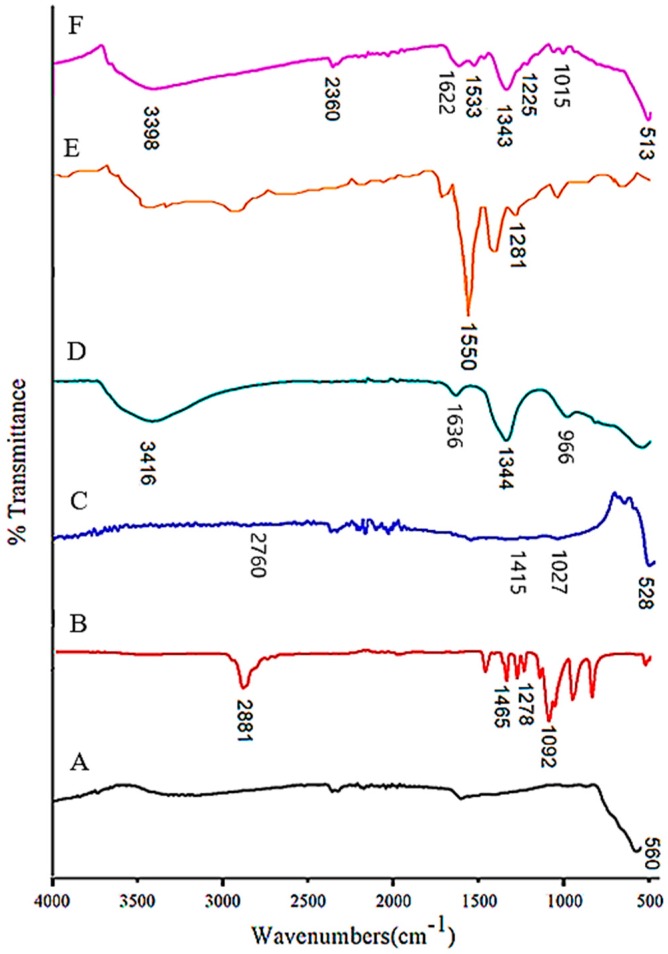
Fourier-transform infrared spectra for (**A**) magnetite nanoparticles; (**B**) pure PVA; (**C**) FPVA; (**D**) pure Mg-Al-LDH; (**E**) 5-fluorouracil; (**F**) core-shell nanoparticles (FPVA-FU-MLDH).

**Figure 4 ijms-20-03764-f004:**
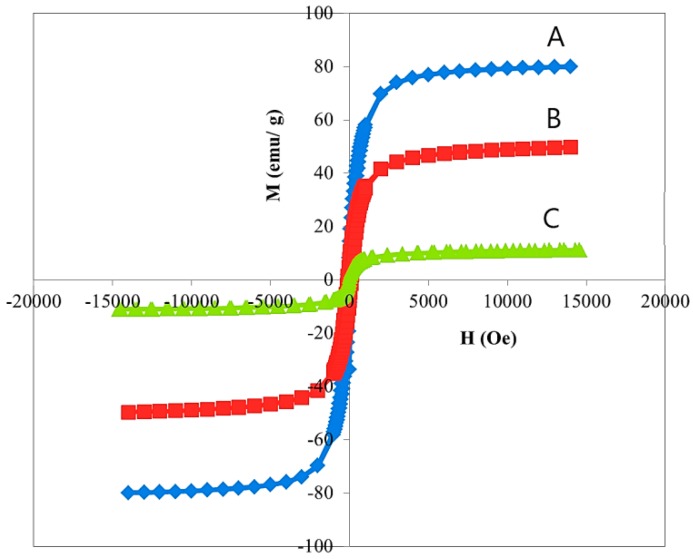
Magnetization curves of (**A**) magnetite nanoparticles; (**B**) magnetite-coated polyvinyl alcohol, (**C**) FPVA-FU-MLDH nanoparticles. Notes: The data are presented in terms of M_s_, mass magnetization (emu/g), versus H, applied magnetic field (O_e_).

**Figure 5 ijms-20-03764-f005:**
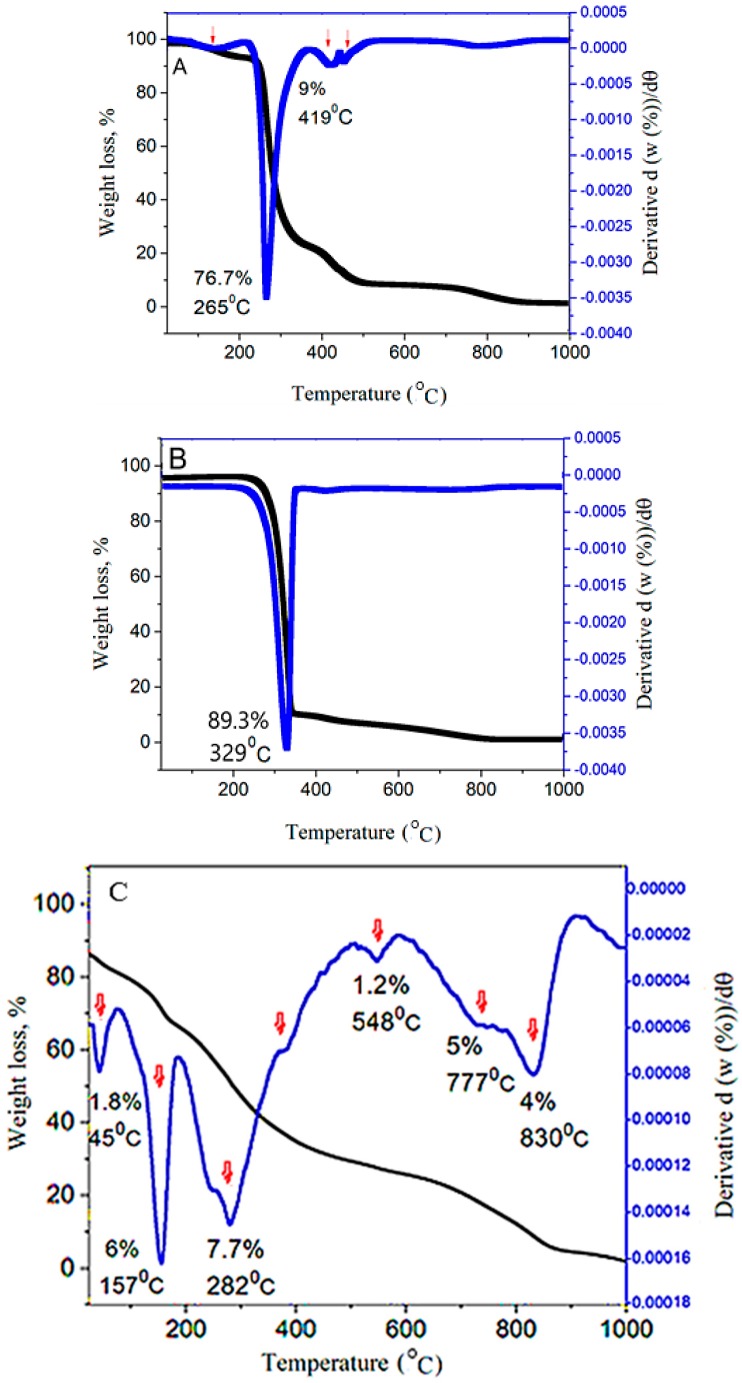
Thermogravimetry analyses of (**A**) polyvinyl alcohol; (**B**) 5-fluorouracil, (**C**) core-shell nanoparticles (FPVA-FU-MLDH). Note: The red arrows point to the weight loss percentages indicated.

**Figure 6 ijms-20-03764-f006:**
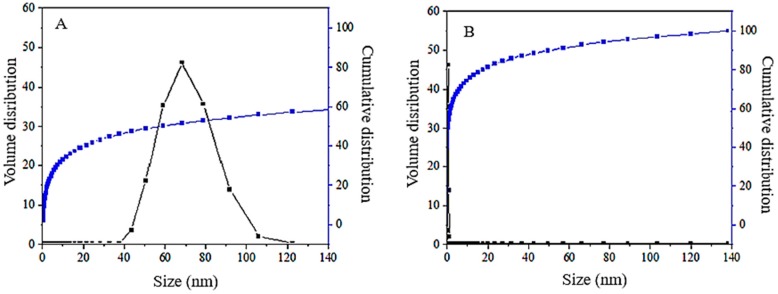
The relative (**A**) and the cumulative (**B**) particle size distribution of core-shell nanoparticles, FPVA-FU-MLDH.

**Figure 7 ijms-20-03764-f007:**
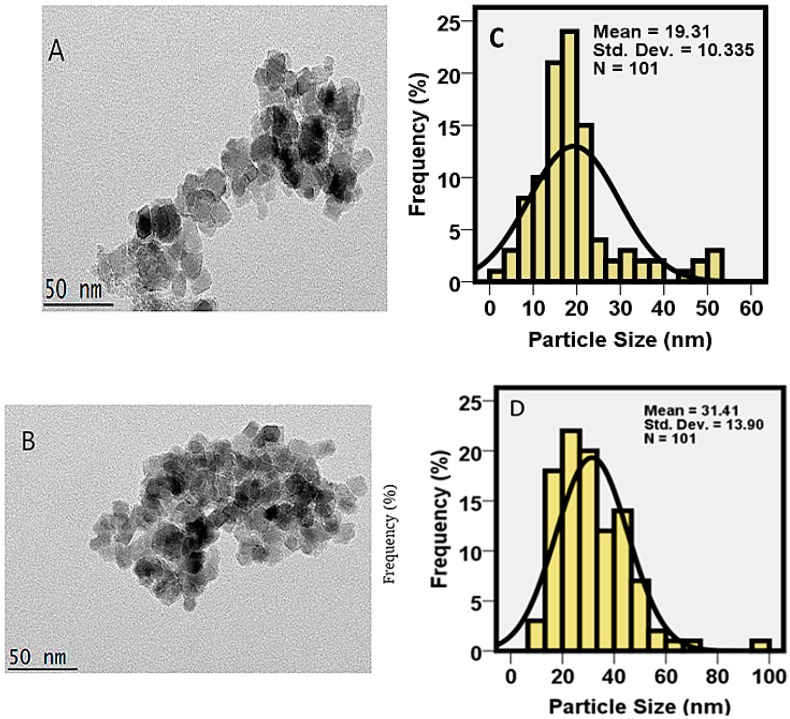
Transmission electron micrographs for (**A**) magnetite nanoparticles; (**B**) magnetite nanoparticles coated with polyvinyl alcohol, co-coated with MLDH, and loaded with 5-fluorouracil; (**C**) particle size distribution of magnetite nanoparticles; (**D**) particle size distribution of FPVA-FU-MLDH core-shell nanoparticles.

**Figure 8 ijms-20-03764-f008:**
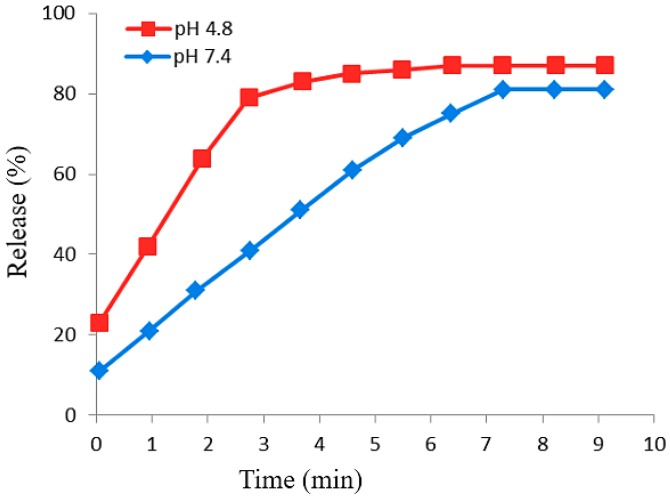
Release profiles of 5-fluorouracil from its physical mixtures into the phosphate-buffered solution at pH 4.8 and pH 7.4.

**Figure 9 ijms-20-03764-f009:**
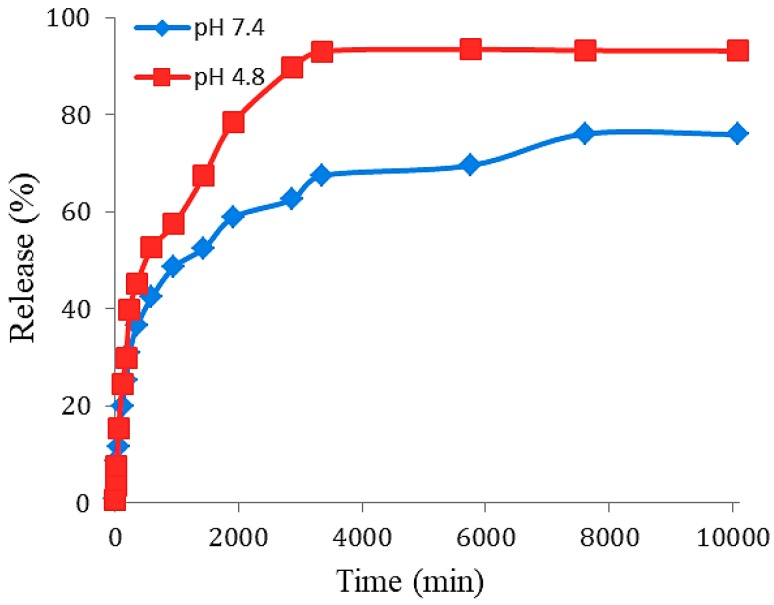
The cumulative release profiles of 5-fluorouracil from its FPVA-FU-MLDH nanoparticles in phosphate-buffered solution at pH 4.8 and pH 7.4.

**Figure 10 ijms-20-03764-f010:**
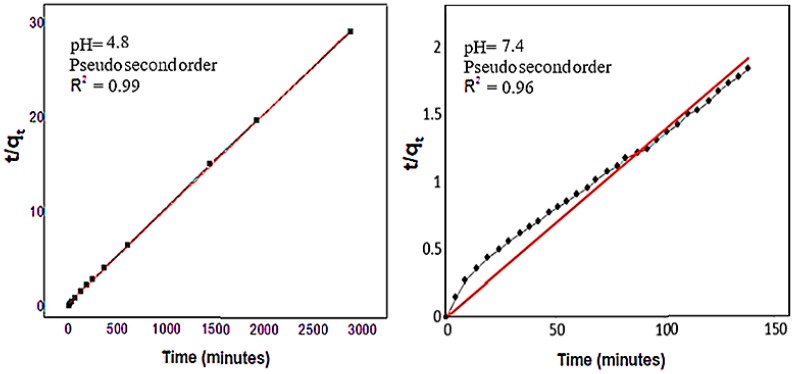
Fitting the data of the release of 5FU from its FPVA-FU-LDH nanoparticles at different solutions at pH 7.4 and pH 4.8 to the pseudo-second-order kinetics. Abbreviations: t, time, q_t_, release at time t.

**Figure 11 ijms-20-03764-f011:**
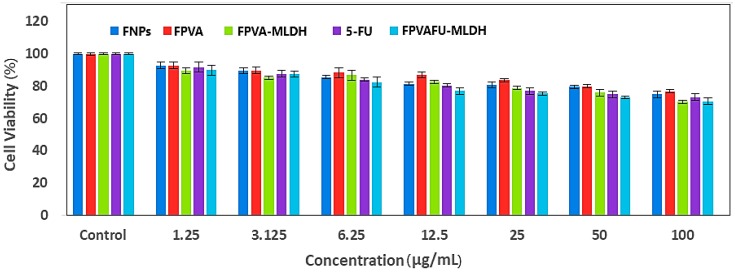
Cytotoxicity assay of magnetite (FNPs), FPVA, FPVA-MLDH, 5FU, and core-shell nanoparticles (FPVA-FU-MLDH) against normal 3T3 fibroblast cells at 72 h of incubation.

**Figure 12 ijms-20-03764-f012:**
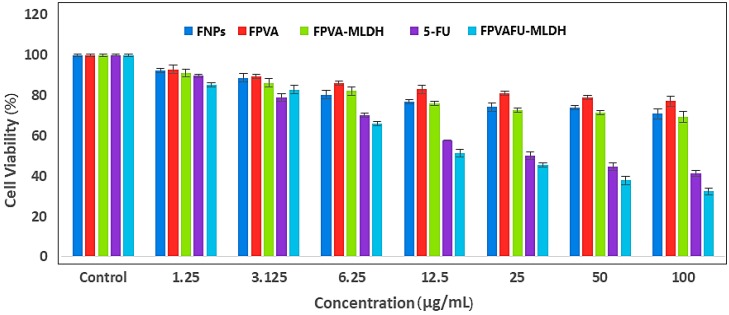
Cytotoxicity assay of magnetite (FNPs); FPVA; FPVA-MLDH; 5FU; FPVA-FU-MLDH against normal HepG2 cells at 72 h incubation.

**Table 1 ijms-20-03764-t001:** Magnetic properties of FNPs, FPVA and FPVA-FU-MLDH nanoparticles.

Samples	M_s_ (emu/g)	M_r_ (emu/g)	H_ci_ (G)
FNPs	80	1.448	11.53
FPVA	49	0.784	12.01
FPVA-FU-MLDH	11	0.208	17.5

**Table 2 ijms-20-03764-t002:** The correlation coefficient, the rate constant and half-life time obtained by fitting the data for the release of 5-fluorouracil from its FPVA-FU-LDH nanoparticles into phosphate-buffered solutions at pH 4.8 and pH 7.4.

pH	Saturation Release	R^2^				
		Pseudo First Order	Pseudo Second Order	Parabolic Diffusion Model	Rate Constant (K) (mg/min)	t_1/2_ (min)
7.4	96%	0. 7902	0.9986	0.5024	3.4 × 10^−5^	54
4.8	99%	0. 9965	0.9996	0.8633	4.4 × 10^−4^	66

**Table 3 ijms-20-03764-t003:** The half-maximal inhibitory concentration (IC_50_) value for magnetite (FNPs), FPVA, FPVA-MLDH, 5FU, and FPVA-FU-MLDH samples tested on 3T3 and HepG2 cell lines.

Cells Type	IC_50_ (μg/mL)				
	FNPs	FPVA	FPVA-MLDH	5-fluorouracil	FPVA-FU-MLDH
3T3	NC	NC	NC	NC	NC
HepG2	NC	NC	NC	32.73	21.53

Note: NC = No cytotoxicity.

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
