# Peer review of "The Impact of Magnesium–Aluminum-Layered Double Hydroxide-Based Polyvinyl Alcohol Coated on Magnetite on the Preparation of Core-Shell Nanoparticles as a Drug Delivery Agent"

_ijms, 2019, doi:10.3390/ijms20153764_

Reviewer 1 Report

This manuscript described the preparation of a new core-shell nanoparticle as the carrier for the drug delivery. The nanocarrier was prepared using magnetic nanoparticle as core, and Mg/Al layered double hydroxides as the shell. The anticancer drug, 5-FU was loaded into LDH and then release at acidic condition (pH 4.8). The anticancer cells efficiency was then investigated. This is an interesting work, could be recommended for publishing after addressing following questions.

1. The introduction section is suggested to be re-organised. It is hard to understand the focus of this research through reading the introduction section.

2. The description and figures of this manuscript are poor. For example, Figure 7 had significant larger labels than other figures, and the font is different.

3. The dynamic size distribution of the prepared nanoparticles is suggested to be added. Any changes of surface charge before and after LDH growth?

4. HRTEM is suggested because LDH coating are not noticed in current TEM images.

5. Some of other magnetic nanoparticles for drug delivery are suggested to be included, such as Materials Letters 2016, 162, 207-210, ACS Applied Nano Materials 2018,  1 (9), 5027-5034, and Nanotechnology 2016, 27 (48), 485702

6. A scheme is suggested to be added to better present the idea of this manuscript.

Author Response

Reviewer   1

NO.

comment

reply

note

1

The   introduction section is suggested to be re-organised. It is hard to

understand   the focus of this research through reading the introduction

section.

Thank you very much for your kind suggestions. We have re-organised   the introduction section by including the drug loading comparison from other   researches.

Done

2

The   description and figures of this manuscript are poor. For example,

Figure   7 had significant larger labels than other figures, and the font is

different.

Thank you very much for your kind comments. We have increased the   quality of the figures and tables where possible

Please see Figure 5C, 7C, 7D and Figure 9.

3

The   dynamic size distribution of the prepared nanoparticles is suggested

to   be added. Any changes of surface charge before and after LDH growth?

Thank you very much for your kind suggestions.

We have added DLS analyses in line 232-243.

Unfortunately, we are unable to include the surface charge data   because we don’t have any sample left.  

4

HRTEM is suggested because LDH   coating are not noticed in current TEM images.

We accept and appreciate your comments. We agreed with your   opinion that HRTEM will give better supporting data to indicate that LDH is   present as the coating material.

For your information, for HRTEM studies we have to send our   sample to other institution and have to wait for sometimes due to long ques.   Therefore we decided that the XRD studies were sufficient enough to indicate   that LDH was present in the resulting material.

In addition to this, we have also replotted the particle size   distribution graphs with more clearer and better way (7C and7D)

5

Some of other magnetic nanoparticles   for drug delivery are suggested to be included, such as :

1) Materials Letters 2016, 162,   207-210

2) ACS Applied Nano Materials   2018, 1 (9), 5027-5034,

3) Nanotechnology 2016, 27 (48),

485702

Thank   you very much for your kind suggestions. This manuscript is compared with your   suggested references.

This   is given in

1)   In line 80-82, 105-108.

2)   In line 43-44, 54-57, 108-111.

3) In line 111-113.

6

A scheme is suggested to be added   to better present the idea of this

manuscript.

Done

Line   118-120

Reviewer 2 Report

Dear authors. 

Please use some newest references for best discussion:

10.1186/s12989-017-0199-z, 

10.3390/toxics7010015, 

10.1080/17435390.2019.1576238. 

Author Response

REVIEWER   2

No.

comment

reply

note

1

Please   use some newest references for best discussion:

1) 10/1186s12989-017-0199-z,

2) 10/3390toxics7010015,

3) 10.1080/17435390.2019.1576238.

Thank you very much for your kind suggestions. We have added some   latest references to improve our discussion as indicated in the notes.

For 1) 10/1186s12989-017-0199-z, 2) 10/3390toxics7010015, and 3)   10.1080/17435390.2019.1576238 we  have   added:

1)   In line 99-102, 2) In line 102-104, and 3) In line 57-59,   respectively.

This manuscript is a resubmission of an earlier submission. The following is a list of the peer review reports and author responses from that submission.

Round  1

Reviewer 1 Report

In this manuscript, magnetite nanoparticles coated by polyethylene glycol (PEG) and co-coated with 5-fluorouracil/ Mg/Al- and Zn/Al-layered double hydroxides were synthesized as the original magnetic drug carrier.

The morphology, magnetic property, and elemental composition are basically measured. However, some important questions should be considered for acceptance.

 1) Authors indicated that the synthesized materials showed superparamagnetic property based on the VSM measurement. However, the mean core diameters of nanoparticles are 28 and 40 nm, which are large and generally show ferromagnetic property.

When the nanoparticles dispersed in liquid, the coercivity and remanence are significantly small and show superparamagnetic behavior despite ferromagnetic nanoparticles in solid state (Journal of Applied Physics 117, 17D713 (2015); doi: 10.1063/1.4914061).

Please show the prepared samples in the VSM measurement in detail. Were the samples liquid and solid? How to prepare the samples?

 2) The saturation magnetizations of nanoparticles were decreased after surface coating. Authors showed that PEG polymer coated on the surface of Fe3O4 nanoparticles could affect the surface magnetic anisotropy and increase surface spins disorientation. Please show the references for the evidence of this discussion.

Is the reduction of the saturation magnetization due to the decrease of magnetic materials such as Fe composed in nanoparticles? The unit of the magnetization was emu/g. What weigh is this?

 3) What is the novelty of this manuscript? In th section of conclusions, authors indicated that the nanocarrier absolutely enhanced the percentage of drug loading with narrow size distribution and smaller particle size by MLDH as the co-coating agent.

In the section 4.3, the percentage of the drug loaded in nanoparticles was shown. If possible, please show the data in more detail and discuss the reason why the percentage of the drug loading was increased. Moreover, please show the percentage of the drug loading of the conventional materials as references.

Authors should show the result and discussion with respect to the novelty of this manuscript in more detail.

 4) Please introduce the other drug loading materials in conventional researches particularly in recent years such as the references shown blow.

Journal of Controlled Release 272 (2018) 145–158; doi: 10.1016/j.jconrel.2017.04.028

ACS Appl. Mater. Interfaces 10 (2018) 12518−12525; doi: 10.1021/acsami.8b02398

 Minor errors

5) Please delete [] in line 48, page 2.

6) Please modify the term from “Recently years” to “In recent years” in line 51, page 2.

Reviewer 2 Report

See attached file.
